# Recent Advances in Nanoparticles-Based Platforms Targeting the PD-1/PD-L1 Pathway for Cancer Treatment

**DOI:** 10.3390/pharmaceutics14081581

**Published:** 2022-07-29

**Authors:** Xin Yu, Chao Fang, Kun Zhang, Chunxia Su

**Affiliations:** 1Department of Oncology, Shanghai Pulmonary Hospital & Thoracic Cancer Institute, Tongji University School of Medicine, Shanghai 200433, China; 1931215@tongji.edu.cn; 2Department of Medical Ultrasound, Shanghai Tenth People’s Hospital, Ultrasound Research and Education Institute, Tongji University School of Medicine, Shanghai 200092, China; max999515@163.com

**Keywords:** PD-1/PD-L1, immune checkpoint, nanoparticles, delivery nanoplatforms, cancer immunotherapy

## Abstract

Immune checkpoint inhibitors (ICIs) targeting the PD-1/PD-L1 axis showed remarkable improvements in overall response and patient survival, which changed the treatment landscape for multiple cancer types. However, the majority of patients receiving ICIs are either non-responders or eventually develop secondary resistance. Meanwhile, immunological homeostasis would be destroyed as T cell functions are activated excessively, leading to immune-related adverse events (irAEs). Clinically, a large number of irAEs caused by ICIs occurred and affected almost every organ system, resulting in the discontinuation or even the termination of the ongoing therapy. Therefore, researchers are exploring methods to overcome the situations of insufficient accumulation of these drugs in tumor sites and severe side effects. PD-1/PD-L1-targeted agents encapsulated in nanoparticles have emerged as novel drug delivery systems for improving the delivery efficacy, enhancing immune response and minimizing side effects in cancer treatment. Nanocarriers targeting the PD-1/PD-L1 axis showed enhanced functionalities and improved the technical weaknesses based on their reduced off-target effects, biocompatible properties, multifunctional potential and biomimetic modifications. Here, we summarize nanoparticles which are designed to directly target the PD-1/PD-L1 axis. We also discuss the combination of anti-PD-1/PD-L1 agents and other therapies using nanomedicine-based treatments and their anticancer effects, safety issues, and future prospects.

## 1. Introduction

A variety of treatments have been developed for different malignancies, including surgery, radiotherapy, chemotherapy, immunotherapy, and others. In recent years, cancer immunotherapy has represented a turning point with the widespread use of immune checkpoint inhibitors (ICIs). Unlike other therapies, immunotherapy aims to boost the immune system and eliminate tumors, ultimately aiding in treating cancers, reducing metastasis, and preventing recurrence. There are a number of checkpoints that have been discovered and targeted, including PD-1/PD-L1, CTLA-4, LAG-3, TIM-3, TIGIT, and so on [1]. Among them, the PD-1/PD-L1 pathway appears to be the most effective and successful target in the drug development pipelines. ICIs targeting PD-1 and PD-L1 have provided durable clinical responses and long-term survival in a subset of patients. However, the majority of patients administered with PD-1/PD-L1 antibodies are either non-responders or eventually develop acquired resistance [2]. Thus, several techniques have been introduced in clinical practice to improve the efficacy of immunotherapy, including the combination of immunotherapy with chemotherapy, radiotherapy, antiangiogenic therapy, and other immunomodulatory drugs [3]. These approaches are designed to impinge on different components of tumor biology to achieve additive or synergistic antitumor activities. However, another concern are more serious side effects in the context of combination therapy.

In order to increase the proportion of patients who would benefit from immunotherapy, novel medications or therapies have been studied and developed that are both effective and low toxicity. There is an urgent need to break the traditional drug design paradigm and optimize the delivery of immunotherapeutics to achieve the precise targeting of tumors and tumor-draining lymph nodes and control the circulation and location of these drugs. To reach this goal, nanomedicine, a formulation of drugs in nanocarriers that are usually smaller than 100 nm, may provide the solutions for increasing both the efficacy and safety of immunotherapy, owing to their site-specific stimulation of tumor-specific immune cells [4]. Several therapeutic nanoparticle (NPs) platforms, such as liposomes, polymers, and albumin NPs have already been utilized for cancer treatment to improve a drug’s solubility and stability, alter systemic exposure, promote tumor infiltrations, and enhance the uptakes of tumor cells. Moreover, many in-depth innovated NPs targeting the immune system are springing up with distinct properties which introduce new complexities in nanomedicine development. Applying these NPs clinically remains a promising strategy to assist in cancer immunotherapy (Figure 1).

This review covers the nanoplatforms targeting the PD-1/PD-L1 pathway, which decrease off-target effects and increase antitumor effects. We also present the basis and rationality of combining PD-1/PD-L1 blockers with other therapies, including chemotherapy, radiotherapy, other checkpoint blockers, and emerging therapies, such as photodynamic therapy, photothermal therapy, and multifunctional NPs. These combination therapeutic strategies exert synergistic effects and achieve tumor inhibition both in situ and at metastatic sites. Finally, given the current gap between immunotherapy and nanotechnology, the perspectives of obstacles and challenges of nanomedicine to be applied in clinical practice are also discussed. The combination strategies of PD-1/PD-L1 blockers with other therapies are depicted in Figure 2.

## 2. Nanoparticles Directly Targeting the PD-1/PD-L1 Pathway

### 2.1. Nanoparticles Delivering Antibodies and Peptides

Despite the outstanding therapeutic effect of PD-1/PD-L1 antibodies, limitations still exist, such as the non-specific delivery of drugs and their limited intratumoral accumulation levels. NP-based ICI therapies could enhance the tumor-specific immune responses and reduce the irAEs to increase the safety profiles. Lim et al. prepared the phenylboronic acid–antibody (pPBA-Ab) nanocomplex by the simple mixing of polymeric phenylboronic acid (pPBA) and anti-PD-L1 antibodies at the desired ratio [5]. The phenylboronic ester was formed by the phenylboronic acid and the diol moiety of the glycosylation site of the antibody. This nanoparticle had an elongated circulation owing to the protective polymer shell and could accumulate in tumor sites by the EPR effect. Additionally, the dissociation of phenylboronic ester and the release of loaded cargo were accelerated in the acidic tumor microenvironments. In vivo studies demonstrated that this nanocomplex showed superior antitumor effects than free antibodies in MC-38 tumor-bearing mouse models. Moreover, Bu et al. created dendrimer–ICI conjugates (G7-αPD-L1) to increase the binding avidity of ICI antibodies by the multivalent binding effect [6]. This system showed enhanced binding strengths to PD-L1 molecules compared with free αPD-L1, which could translate into the efficiency and antitumor effects both in vitro and in vivo. The efficacy of ICIs was unsatisfactory in intracranial tumors in part because of the insufficient drug penetration across the blood–brain barrier (BBB). As a benzamide analogue, p-hydroxybenzoic acid (pHA) displayed great capabilities to cross the BBB. A study designed a conjugate of targeting moiety pHA and anti-PD-L1 for crossing the BBB mediated by dopamine receptors [7]. Of note, pHA-αPD-L1 complex exhibited enhanced accumulation in the brain and glioma after being given intravenously. In vivo study demonstrated that pHA-αPD-L1 could prolong the survival of orthotopic glioblastoma mice models by activating T cell immune response and relieving immunosuppressive TME.

Ferritin-based nanocages with multiple loadings could awaken the host immune system and provoke a strong and durable response in cancer treatment, which are attractive as nanocarriers [8]. A study developed PD-1-decorated nanocages (PdNCs) by surface engineering, which could remarkably increase the binding affinity of the ligands and provide enhanced antagonistic efficiency. With a desirable nanocage size of about 20 nm, PdNCs could rapidly be drained and accumulated into the tumor-draining lymph nodes (TDLNs). In vivo studies showed that the PdNCs treatment induced the maturation of the DC and the tumor-specific T cell responses both in TDLNs and TMEs. These behaviors can inhibit tumor growth and induce tumor eradication in some tumor-bearing mice [9]. 5-Aza-20-deoxycytidine (DAC) functions as an epigenetic agent to block de novo DNA methylation in activated PD-1+CD8+ TILs and can help to boost antitumor immunity in immunotherapy. Hu et al. designed polyethylene glycol-poly(ε-caprolactone) (PEG-PCL) NPs with anti-PD-1 antibody nivolumab linked in the polymer’s core for targeting TILs overexpressing PD-1 [10]. PEG-PCL polymers improved IFN-γ secretion and antitumor efficacy of nivolumab in vitro studies. Thus, the delivery of DAC reversed the function of PD-1+CD8+ TILs and enhanced the efficacy of immunotherapy, suggesting this NP may become a potential tool for delivering epigenetic drugs.

Foreign substances used as nanocarriers are easily detected and cleared by the immune system, making them difficult to apply clinically. To overcome these obstacles, biomimetic nanomedicines were developed using biofilm engineering and served as a delivery system to help drugs evade immune clearance and prolong circulation. Platelets are considered as immune ‘cells’ that assist and modulate many inflammatory reactions. A previous study revealed that platelets conjugated with anti-PDL1 can reduce the risk of tumor recurrence and metastasis after surgery [11]. Furthermore, Hu et al. described a hematopoietic stem cells (HSC)-platelet cellular combination delivery system. HSCs conjugated with platelets decorated with anti-PD-1 can facilitate the transport of this complex to the bone marrow and release anti-PD-1 when administered intravenously into leukemia-bearing mice. Immune response was enhanced as the number of activating T cells and cytokines increased, which significantly prolonged the survival time of the mouse models [12]. In addition to the biomimetic nanomedicines, Zhang et al. also reported that engineered cellular nanovesicles with PD-1 receptors expression on their surfaces enhance antitumor responses by disrupting the PD-1/PD-L1 immune inhibitory axis [13].

Most checkpoint inhibitor antibodies are designed and given intravenously because peptides present a big challenge regarding oral administration. Using molecular dynamics simulations combined with in silico mutagenesis, the enzymatic resistant peptide OPBP-1 (Oral PD-L1 Binding Peptide 1) was obtained based on the structure of DPPA-1 to improve the blocking efficacy [14]. Li et al. used trimethyl chitosan (TMC) hydrogel as carriers to load peptide OPBP-1 in order to enhance the oral bioavailability and improve the gastrointestinal mucosal adhesion and permeation. Importantly, OPBP-1@TMC significantly inhibits tumor growth in the CT26 model compared to the control group, enhancing the function of CD8+ T cells [15]. These results provide us with a new prospect for developing oral delivery systems in the future.

### 2.2. Nanoparticles Delivering RNA

RNA interference (RNAi) is a conserved cellular pathway of post-transcriptional gene regulation, which includes endogenous microRNA (miRNA) and short double-stranded RNAs called small interfering RNA (siRNA). siRNA can be designed to knock down any genes, which expands the druggable targets of the human genome extremely. For cancer treatment, siRNA could be used to inhibit the expression of genes related to cell survival, proliferation, and cell cycle progression, providing an encouraging opportunity for drug design [16]. Thus, it is possible to use siRNA for targeting PD-L1 and reduce the de novo expression of PD-L1 proteins in cancer cells. However, naked siRNA is prone to degradation, too large, and too negatively charged to pass through the cell membrane. In order to implement it in clinical practice, effective and safe delivery systems have to be developed [17].

Kwak et al. designed a complex to deliver siRNA with a polymeric carrier (“pd”) consisting of disulfide-cross-linked polyethylenimine (CLPEI) and dermatan sulfate (DS). This siRNA/pd reduced the expression of PD-L1 and attenuated the expression of cancer-related genes in B16F10 cells. Both C57BL/6 mice and immune-compromised nude mice models showed tumor growth suppression. They also observed a strong correlation between PD-L1 and p-S6k, a marker of mTOR pathway activation in tumors, indicating siRNA/pd could act on both immune checkpoint and the tumorigenesis signaling pathway [18]. In addition, further research intended to deliver the PD-L1 siRNA by folic acid (FA)–functionalized polyethylenimine (PEI) polymers. These complexes increased the uptake of siRNA into SKOV-3 epithelial ovarian cancer cells, which express folate receptors and decreased monocyte uptake, resulting in about 40–50% PD-L1 protein knockout. Notably, compared with siRNA controls, SKOV-3 cells treated with (PEG)-FA/PD-L1 siRNA were twice more sensitive to T cell killing [19]. These results highlight the modified PEI for potentially being used as gene carriers.

One of the key technological problems for delivering siRNA into cells is overcoming the lipid bilayer, which means escaping from the endosome and reaching the cytoplasm to silence gene expression [20]. In order to achieve efficient endosome escape, Li et al. formed a pH-sensitive copolymer siRNA-loaded PDDT nanomicelles (PDDT-Ms/siRNA) and investigated in vitro and in vivo performance [21]. In an endosome pH environment (6.5–6.8), PDDT-Ms/siRNA could rapidly disassemble, resulting in the release of the internalized siRNA to cytoplasm space to silence gene expression using a lysosomotropic agent (chloroquine) and the endosomal transporting inhibitor (bafilomycin A1). Moreover, PDDT-Ms/siPLK1 reduced tumor growth in HepG2-xenograft and patient-derived xenograft models and restored immunological surveillance in CT-26-xenograft mice models with a good safety profile. This research optimized the design of polymer complex for the endosomal escape of siRNA in cancer treatment.

The above studies showed that the development of genetic intervention techniques to silence PD-1/PD-L1 genes provides an exciting opportunity as a substitute for checkpoint inhibitor antibodies. Apart from siRNA, small hairpin RNA of PD-L1 (shPD-L1) could also be applied to silence gene expression. Guan et al. designed shPD-L1-loaded ultrasensitive pH-triggered nanoparticles [22]. Hyaluronidase (HAase) was used for degrading hyaluronic acid of the tumor tissues to enhance the penetration of the nanoparticles. In vivo experiments demonstrated that the combination of HAase and shPD-L1@NPs achieved increased PD-L1 gene silence and a better antitumor effect than shPD-L1@NPs alone in the melanoma mice model. Thus, this combination approach has great potential to be utilized in the future.

## 3. Nanoparticles for Combining PD-1/PD-L1 Blocking and Other Therapies

### 3.1. Nanoparticles Combining PD-1/PD-L1 Blocking and Chemotherapy

A rational combination of immunotherapy and chemotherapy can influence different elements of tumor biology to achieve additive or synergistic antitumor effects. The two main ways in which chemotherapy promotes tumor immunity are by inducing immunogenic cell death as part of its intended therapeutic effect, and by disrupting strategies used by tumors to evade immune recognition [23,24]. Although this combination regimen had shown superior to chemotherapy or immunotherapy alone in many cancer types, management of treatment-related adverse events was always a tough question. Thus, it is essential to develop novel delivery systems to achieve the co-delivery of the two categories of drugs. Duan et al. developed a self-assembled core-shell nanoparticle (OxPt/DHA) in which the nano coordination polymers of Zn and OxPt prodrugs were coated in lipid bilayers containing cholesterol-DHA conjugate (chol-DHA). OxPt and DHA had strong synergic effects in generating reactive oxygen species (ROS) and anticancer functions. Furthermore, the addition of anti-PD-1 antibodies with OxPt/DHA therapy could awaken innate and adaptive immune responses and eradicate tumors of murine colorectal cancer [25]. Kinoh et al. applied pH-sensitive epirubicin-loaded micellar nanomedicines to synergy with anti-PD-1 antibodies against both PTEN-positive and PTEN-negative orthotopic glioblastoma. The combination therapy could transform the cold TME into a hot condition with the high infiltration of anticancer immune cells and the eradication of immune suppressive cells [26]. A combination of leucovorin (LV) and fluorouracil (FU) with oxaliplatin (FOLFOX) has been known as the standard first-line therapy for advanced colorectal cancer for decades [27]. In order to improve the clinical benefits of the FOLFOX regimen, Guo et al. formed a nanoparticle using the active form of oxaliplatin and folinic acid, which were encapsulated into an aminoethyl anisamide-targeted PEGylated lipid NPs [28]. As a result, Nano-Folox achieved stronger chemo-immunotherapeutic responses compared to FOLFOX. Anti-PD-L1 antibodies further enhanced the efficacy of Nano-Folox in reducing liver metastases in an orthotopic CRC mouse model.

Recent research showed that miR200C could inhibit PD-L1 expression and improve the susceptibility of cancer cells to chemotherapy agents. Phung et al. constructed dual drug-loaded nanoparticles delivering doxorubicin (DOX) and miR-200c (DOX/miR-NPs) [29]. Folic acid-modified NPs increased the uptake by tumor cells in vitro and the accumulation in the TME in vivo. Notably, DOX/miR-NPs significantly inhibited PD-L1 expression and induced immunogenic cell death both in vitro and in vivo. These results revealed that miR200c could facilitate the efficacy of chemotherapy drugs in cancer treatment. Peptide drugs targeting the PD-1/PD-L1 pathway have emerged as a promising therapy for cancer. Phosphatidylinositol 3-kinase (PI3K) is known to influence the balance between M1/M2 macrophage polarization to regulate the immune response. Song et al. designed an albumin NP (Nano-PI) encapsulating paclitaxel (PTX) and the PI3K inhibitor (IPI-549) [30]. The combination of anti-PD-1 with Nano-PI remodeled the TME in two breast cancer mice models and induced long-term tumor remission. Mechanically, PTX combined with IPI-59 enhanced the M1 repolarization, increased immune-effector cells and decreased immune-suppressive cells, while Nano-PI facilitated the delivery of the two drugs to lymph nodes and tumor sites. This strategy represented a potential candidate for clinical practice. Li et al. developed a self-assembled nanovehicle (SNV) from the immunogenic cell death-inducing copolymer and PD-L1 blocking copolymer to elicit mitochondrial-targeted immunogenic cell death induction and PD-L1 blocking [31]. It has the properties of long-circulating, better accumulation into tumors and mitochondrial targeting. In vivo studies demonstrated the remarkably immune response and antitumor activity in both B16F10 and 4T1 tumors.

Previous work showed that the ferritin cage could fuse targeting peptides on its surface to form peptide bundles on nanocages (PBNCs) that synergistically increase binding affinity [32]. Jeon et al. designed a ferritin nanocage displaying 24 PD-L1 binding peptides (PpNF) on the surface [33]. Moreover, encapsulated with doxorubicin, PpNF showed more effective antitumor activity than free anti-PD-L1 antibody in tumor-bearing mice. These nanocages were potential to serve as nanoplatforms for combining immunotherapy and chemotherapy.

Membrane nanovesicles (NVs) had advantages in delivering drugs, such as high flexibility, ease for surface manipulation, and great biocompatibility and biodegradability. Using genetic engineering methods, antibodies could be linked to the exterior of cell-membrane-derived nanovesicles. The antibodies maintained their biological activity and efficiently delivered drugs to specific tumor cells [34]. Li et al. prepared cluster of differentiation 64 (CD64) as fragment crystalline (Fc) catchers and to be overexpressed in membrane NVs for PD-L1 antibody delivering (CD64-NVs-aPD-L1) [35]. Meanwhile, the chemotherapy agent cyclophosphamide (CP) was loaded into this NP to enhance antitumor effects by unleashing the effect of CD8+ T cells and restraining Tregs activity. In the mouse melanoma model, CD64-NVs-aPD-L1 could intensively suppress the tumor growth and improved the survival time without obvious changes in body weight. In summary, membrane-derived NVs could achieve both checkpoint blockades and chemotherapy for combined cancer immunotherapy.

### 3.2. Nanoparticles Combining PD-1/PD-L1 Blocking and Radiotherapy

As a hallmark of GBM, tumor-associated myeloid cells (TAMCs) are key factors of immunosuppression in the TME and could account for up to 50% of the tumor mass, which impedes the efficacy of immunotherapy or other traditional therapies. As PD-L1 molecules are highly expressed on glioma-associated TAMCs, Zhang et al. designed and reported a lipid nanoparticle (LNP) decorated with anti-PD-L1 antibodies on its surface [36]. This platform (αPD-L1-LNP) was further encapsulated with a cyclin-dependent kinase (CDK) inhibitor dinaciclib to induce the depletion of TAMCs and impair their immunosuppressive roles. Notably, the targeting efficiency of αPD-L1-LNP/Dina was strengthened with additional radiotherapy (RT) due to the RT-induced PD-L1 upregulation on TAMCs. Combining RT with αPD-L1-LNP/Dina resulted in the prolonged survival of two syngeneic glioma mice models of CT2A and GL261. This phenomenon was also validated towards human TAMCs from GBM patients. Thus, this research explored a potential approach to improve the clinical outcomes of patients with GBM and needed to be further verified. In another study, Erel-Akbaba et al. designed a solid lipid nanoparticle (SLN) conjugating with a cyclic peptide iRGD to deliver siRNAs against PD-L1 and EGFR for GBM [37]. They found that low-dose radiation facilitated SLN to locate in the tumor site, leading to the down-regulation of PD-L1 and EGFR expression. RT followed by the administration of SLN resulted in a decrease in tumor growth based on bioluminescence imaging. Overall, this work illustrated an approach for using RT to increase NPs uptake and effectively target the EGFR and PD-L1 pathways in GBM, which could be probably extended to other malignancies.

Confirmation and intensity-modulated RT have been extensively investigated in the past few decades. Radiation enhancers can not only decrease the dose of X-rays but also precisely target the tumor cells to reduce side effects [38]. Ni et al. developed two porous Hf-based metal-organic frameworks (nMOFs), Hf6-DBA and Hf12-DBA nMOFs, that significantly outperformed HfO2 to enhance the effects of the X-ray RT [39]. The combination of nMOF-mediated RT with PD-L1 blockade effectively eliminated the primary tumor and distant tumor via abscopal effects. The advantages of nMOFs included tumor-targeting properties to minimize the irradiation dose and maintain sufficient ionizing damage to the tumor cells. This work proved the feasibility of using nMOFs as an assisted therapy for immunotherapy.

### 3.3. Nanoparticles Combining PD-1/PD-L1 Blocking and Other Immune Checkpoint Blockers

At present, there are many other immune checkpoints been discovered and served as drug targets, such as CTLA-4, TIM-3, LAG-3, TIGIT, VISTA, etc. As some drugs enter clinical trials and daily practice, PD-1/PD-L1 inhibitors combined with other checkpoint blockers have shown significant immune response and tumor shrinkage. In order to achieve better synergistic effects, nanoparticles for the combination of immunotherapy approaches are urged to be investigated. A self-degradable microneedle (MN) patch composed of hyaluronic acid and dextran nanoparticles that loaded anti-PD-1 and glucose oxidase was developed [40]. In vivo study revealed that the efficacy of the MN patch in inhibiting tumor growth was better than the intratumoral injection of free anti-PD-1. In addition, the combination of anti-CTLA4 and anti-PD-1 antibodies’ co-delivery by MNs showed a remarkable synergistic effect in comparison to either acting individually in the B16F10 melanoma mice model. Of mice receiving the combination regimen, 70% achieved complete control of the tumor and reached disease-free survival in 60 days. Galstyan et al. developed targeted nanoscale immunoconjugates (NICs) using a versatile drug carrier poly (β-L-malic acid) (PMLA) and covalently attached with αPD-L1 and αCTLA-4 for systemic delivery across the blood–brain barrier and glioma therapy [41]. The distribution of drugs in tumor sites was examined using two vascular labeling methods and tumor parenchyma showed significantly more NIC-attached antibodies than the free antibodies. NICs treatment also leads to increased immune-activating cells in tumor areas and favorable prognoses in glioma-bearing mice. Notably, the NICs combination showed the highest survival of mice compared to single NICs or free antibodies, demonstrating the superior efficacy of dual checkpoint inhibition.

It is generally agreed that acidic pH is a major attribute of TME that enhances tumor immune evasion. Jin et al. found that an acidic pH environment could upregulate co-inhibitory checkpoint receptors like PD-1, TIM-3, LAG-3, and TIGIT, and inhibit Akt/mTOR activation in memory CD8+ T cells [42]. Additionally, extracellular tumor acidity increased the suppressive function of the TIGIT-CD155 axis. Therefore, they used Pluronic F-127 (a NaHCO3 releasing carrier) to alleviate the extracellular acidic pH environment. In vivo experiments showed that pH modulating injectable gel (pHe-MIG) transformed the immuno-suppressive TME to a favorable condition and the combination of pHe-MIG with PD-1 and TIGIT inhibitors boosted the immune response and synergistically improved the antitumor effects, which could be a novel method for designing immunotherapies. Huang et al. designed a liposome-based photothermal therapy (PTT) nanoparticle through the self-assembling injectable lipids and photothermal agent indocyanine green (ICG) [43]. The obtained nanoparticle could efficiently eradicate the tumor in CT26 and MC38 mice models. Moreover, the distant tumor growth correlated with the upregulation of immune checkpoints like PD-1 and TIM-3 after PTT therapy. Dual PD-1 and TIM-3 blockade combination generated a systemic response and significantly inhibited tumor growth both in primary and distant sites.

### 3.4. Nanoparticles Combining PD-1/PD-L1 Blocking and Photodynamic Therapy (PDT)

PDT is a non-invasive and modern form of therapy, which is based on the application of photosensitizers (PS). Mechanistically, PS molecules are activated by the light with an appropriate wavelength, aiming to destroy the target cells in the pathological tissues [44].

PDT could be divided into type I PDT and type II PDT based on their photochemical reaction processes, generally, type I PDT occurs in hypoxic environments and type II PDT dominates in oxygenated conditions [45]. With the development of nanotechnology, nanoparticles combined with photosensitizers can improve the efficiency and selectivity of photodynamic therapy and reduce adverse events as well. Here, we discuss the rationale for combining PD-1/PD-L1 blockades and PDT to expand the application for immunotherapy in cancer.

Feng et al. synthesized the acid-responsive polygalactose-co-polycinnamaldehyde polyprodrug (PGCA) for self-assembling into NPs that deliver PA (PGCA@PA NPs) [46]. ROS level was increased by the combination of cinnamaldehyde (CA) and photosensitizer pheophorbide A (PA) with light irradiation in cancer cells. Moreover, when combined with anti-PD-1 therapy, PGCA@PA NPs significantly promoted T cell infiltration and boost immune response in melanoma. In another study, Duan et al. made Zn-pyrophosphate (ZnP) nanoparticles encapsulating the photosensitizer pyrolipid (ZnP@pyro) [47]. PDT made nanoparticles sensitive to PD-L1 checkpoint inhibition, resulting in the eradication of both primary and untreated distant tumors in 4T1 and TUBO mouse models.

The efficacy of PDT in cancer treatment is usually limited because of its hypoxia resistance. Based on nanoscale metal–organic frameworks (nMOFs), Lan et al. designed Fe-TBP as a photosensitizer to sensitize PDT in hypoxic conditions [39]. Fe-TBP was composed of iron-oxo clusters and porphyrin ligands, which exhibited effective cellular uptakes. Meanwhile, Fe-TBP-mediated PDT could enhance the accumulation of cytotoxic T cells in TME. The combination of Fe-TBP and anti-PD-L1 treatment resulted in the >90% regression of tumors in the colorectal mouse model. In another study, researchers delivered a hypoxia-activated prodrug (AQ4N) in order to amplify the effect of PDT [48]. They utilized a water-soluble phthalocyanine derivative (PcN4) that bound to endogenous albumin dimers to become complexes with tumor-targeting properties and improved the activity of PDT. The additional combination therapy of PD-L1/PD-1 blockades with PcN4/AQ4N achieved an enhanced abscopal effect in metastatic and distant tumors. Zhang et al. developed a biomimetic nanoemulsion PHD@PM camouflaged with cell membrane expressing PD-1 for synergistic PDT in breast cancer [49]. Perfluorotributylamine (PFTBA) and sinoporphyrin sodium (DVDMS) were encapsulated into the nanovesicles. The perfluorocarbon could transport oxygen and serve as an oxygen source in a hypoxic environment [50]. The co-delivery of a PD-1 protein and a photosensitizer achieved a synergistic effect of photodynamic immunotherapy, which completely inhibited both primary and distant 4T1 tumors in vivo.

Recently, researchers revealed that metformin could reduce hypoxic activation by inhibiting mitochondrial complex I [51]. Moreover, it was reported to increase cytotoxic T cell activity by downregulating the PD-L1 expression on the tumor cell membrane [52]. Thus, metformin could be a competitive candidate drug for photodynamic immunotherapy. Xiong et al. constructed a nanoplatform with liposomes loading metformin and IR775 (IR775@Met@Lip) [53]. This nanoparticle could enhance ROS production and downregulate PD-L1 expression. These benefits effectively reversed Y cell exhaustion and alleviated tumor hypoxia. Similarly, Yang et al. designed PEG-PCL liposomes loading photosensitizers IR780 and metformin [54]. These results indicated that metformin in combination with PDT had great potential to become an effective cancer treatment modality.

Melanoma is one of the most aggressive skin malignancies affecting humans. Wang et al. designed MMP-2-sensitive nanoparticles (S-aPDL1/ICG@NP) integrating anti-PDL1 and photosensitizer indocyanine green (ICG) [55]. Combining anti-PD-L1 nanoparticles with laser irradiation enabled the photosensitizer to induce ROS to release and enhance the efficacy of PD-L1 blockade therapy. In order to overcome the light penetration problem in melanoma, Huang et al. used chlorin e6 (Ce6) as both photosensitizer and sonosensitizer to produce ROS and developed a lipid (LP)-based micellar containing both Ce6 and the anti-PD-L1 antibody (PEG-CDM-aPD-L1/Ce6) [56]. Sonodynamic therapy and immunotherapy facilitated the accumulation and activation of TILs and boosted antitumor response for long-term survival in the mouse melanoma models. This method demonstrated great promise for immuno-sonodynamic treatment and could be extended to pigmented and deep-seated tumors.

Successful PDT based on image guidance could improve the efficacy of cancer treatment. Currently, the techniques of co-delivering the photosensitizer and image contrast agents produce an inconsistency between the release profiles of PDT and image agents. Xu et al. created a nanocomplex called Chloringlobulin using the photosensitizer Chlorin e6 (Ce6) bounding to immunoglobulin G (IgG) [57]. Utilizing the αPD-L1 to prepare αPD-L1 Chloringlobulin, they treated orthotopic glioma-bearing mice with a combination of image-guided surgery, PD-L1 and PDT therapy, and fluorescence image-guided PDT combined with a PD-L1 and CTLA-4 dual blockade, which was used to treat colon cancer. These combinational treatment strategies prolonged survival and elicited a long-term memory response, which provides great potential for clinical practice.

### 3.5. Nanoparticles Combining PD-1/PD-L1 Blocking and Photothermal Therapy (PTT)

PTT utilizes photothermal transduction agents (PTAs) that convert the energy into heat to trigger the thermal ablation of tumors. Compared with other therapies, PTT provides the advantages of the precise targeting of the tumors with adjustable laser irradiation, which avoids severe adjacent healthy tissue damage [58]. Although it is an effective and non-invasive method capable of treating tumors, local recurrence and metastasis would probably happen because of uneven heating and poor immune response. Immunotherapy could elicit a strong immune response to effectively treat a wild variety of cancers. Thus, combining PTT with immunotherapy to generate potential synergistic effects is of great interest to researchers.

Near-infrared (NIR) light-triggered inorganic materials are applied extensively, which show favorable absorbance abilities, great photothermal conversion properties, and excellent photo-stabilities [59]. Zhang et al. synthesized multifunctional NPs loading anti-PD1, perfluorobutane (PFP), and iron oxide (GOP@aPD1) [60]. In vivo, GOP@aPD1 NPs were intravenously injected into B16F10 melanoma-bearing mice, which achieved the synergistic antitumor efficacy mediated with PTT duo to the efficient delivery of anti-PD1 and increased CD8+ T cells accumulation in tumor sites. Of note, GOP@aPD1-PTT-treated mice showed desirable tumor regression compared with anti-PD1 immunotherapy. This study verifies the novel strategy of GOP@aPD1-based PTT, which provides the rationale for combining immunotherapy with photothermal treatment. Different from conventional PTT, which directly kills tumor cells, Huang et al. used mild PTT (MPTT) as a regulatory mechanism for tumor immunotherapy [61]. MPTT can alter the TME by activating the systemic immune response, increasing tumor T cell infiltration, and turning ‘cold’ tumors into ‘hot’ tumors, thereby enhancing the effect of anti-PD-L1 antibody on immune checkpoint blockade. Through the thermal effect of IR820 induced by NIR, the reversible phase transition of the packaging aPD-L1 lipogel (LG) was regulated to achieve the controlled release of aPD-L1 and increase the recruitment of TILs to boost T-cell activity in vivo.

Large-pore mesoporous silica nanoparticles (MSNs) are attracting more attention and interest because of their high biodegradability, large-pore structure, and easy-modified properties [62]. Zhang et al. designed a multifunctional nanoplatform for photoacoustic (PA) and ultrasound (US) photothermal combined immunotherapy [63]. The nanocomposites had a copper sulfide (CuS) core and mesoporous silica shell loading perfluoropentane (PFP) and possessed excellent biocompatibility, PA/US imaging, and strong PTT effect irradicated by 808 nm laser, which indicated a potential application of molecular classification, diagnosis, and treatment of breast cancer. More importantly, in vivo experiments showed that nanoplatform-mediated PTT combined with anti-PD-1 blockade was capable of obliterating primary tumors and inhibiting metastatic tumors. In another study, Chen et al. developed a resembled nanoplatform based on dendritic large-pore MSNs with the ability of photothermal and immune remodeling for treating triple-negative breast cancer (TNBC) [64]. The CuS nanoparticles with great photothermal conversion efficacy were in situ deposited in the pores of MSNs and resiquimod (R848) were encapsulated simultaneously as immune adjuvants. On the surfaces were homogenous cancer cell membranes decorated with anti-PD-1 peptide AUNP-12. The obtained AM@DLMSN@CuS/R848 exhibited high targeting ability and tumor ablation with 980 nm laser irradiation. The photothermal effect caused tumor antigens and R848 releasing, and AUNP-12 was also dissociated via the cleavage of the benzoic-imine bond in the weakly acidic TME, which synergistically boosted immune response and vaccine-like functions to prevent TNBC recurrence and metastasis. The two studies provide an insight into the intelligent nanoplatforms to ameliorate clinical outcomes in metastatic TNBC.

In recent years, cancer-starvation therapy through depleting nutrients or blocking blood supply has been wildly investigated for cancer treatment. Glucose oxidase (GOx) is an endogenous oxidoreductase that catalyzes the oxidization of glucose, which could be employed as a potential strategy [65]. Wang et al. developed a three-in-one cascade nanozyme for glucose consumption, oxygen supply, and photothermal conversion [66]. They utilized double-layered ZIF-8@ZIF-67 to synthesize the Cu-doped cobalt oxide and porous carbon nanocomposites (CuCo(O)@PCNs) and loaded GOx to form the CuCo(O)/GOx@PCNs hybrid-nanozyme. The porous nanocarbon had a photothermal conversion efficiency of 40.04% to eliminate the primary tumor, and the starvation therapy effect was further enhanced by oxygen supply and PTT. Moreover, CuCo(O)/GOx@PCNs can induce antitumor response alone and trigger more effective immunity when combined with α-PD-1 in 4T1-bearing mice. Interestingly, the effect of CuCo(O)/GOx@PCNs seemed to be similar to nanozyme and α-PD-1 combination therapy. This work may provide new ideas for applying the ZIFs-derived CuCo(O)/GOx@PCNs nanozymes as an alternative option to ICIs.

Two-dimensional (2D) nanomaterials have attracted much interest because of their unique structures, great physicochemical properties, and large surface areas. Taking advantage of their unique optical properties, 2D nanomaterials, such as black phosphorus, graphene sheets, and transition metal dichalcogenides, can be utilized for PTT [67,68]. Yan et al. designed IDO inhibitor (IDOi)-loaded reduced graphene oxide-based nanoparticles (IDOi/rGO) to kill cancer cells irradiated by a laser. Meanwhile, the immune response was triggered synergistically by PD-L1 blockade and IDO inhibition, including the enhancement of immuno-activating TILs, such as CD8+ T cells, CD4+ T cells, NK cells, and CD45+ leukocytes, and the inhibition of immunosuppressive Tregs. The combinations of PTT, IDO inhibitor, and PD-L1 blockade could effectively inhibit the growth of both irradiated and non-irradicated tumors [69]. As representative metal phosphorous trichalcogenides (MPX3), FePS3 nanosheets were reported to serve as a 2D ‘all-in-one’ theranostic nanoplatform for cancer treatment [70]. Fang et al. reported a nanoparticle with CT26 cancer cell membrane (CCM) as the outer shell and the FePSe3-based theranostic agent FePSe3@APP@CCM encapsulated with anti-PD-1 peptide as the inner element [71]. Under laser excitation, FePSe3@APP@CCM induced effective tumor ablation by PTT and triggered the immune responses. Moreover, anti-PD-1 peptide blocking the PD-1/PD-L1 axis further activated T cells and raised strong anticancer immunity. Both in vitro and in vivo models demonstrated the synergistic antitumor effects and immune response of PTT and anti-PD-1-targeted immunotherapy.

Black phosphorus quantum dots (BPQD) have the properties of large surface area, ultrahigh charge mobility, anisotropic structure, and enhanced optical absorption, which gained much attention in recent years [72]. Ye et al. used the BPQD-CCNVs coated with the tumor cell membrane and encapsulated into a hydrogel containing GM-CSF and LPS [73]. The hypodermic injection of BPQD-CCNVs caused the constant release of GM-CSF, which recruited dendritic cells to capture antigens coming from the tumor. LPS and irradiation triggered the activation and expansion of DCs, which migrated to the draining lymph nodes for antigen presentation. Moreover, the addition of the PD-1 antibody significantly increased the clearance of surgical residual sites by enhancing CD8+ T cell functions. In another study, Liang et al. prepared a BPQD coated with erythrocyte membranes (RMs) and formed a BPQD-RM nanovesicle (BPQD-RMNV) formulation [74]. Combined with anti-PD-1 treatment, a basal-like breast tumor growth in vivo was significantly inhibited. Hence, these works may provide a promising strategy for the development of BPQD in the era of immunotherapy.

Organic PTT agents are used as alternatives to classical inorganic nanomaterials over the past decade [59]. Other than involving PTT to eliminate tumors, Huang et al. loaded the TGF--β inhibitor SB-505124 and photosensitizer IR780 into nanoliposomes (Nano-IR-SB@Lip) [75]. The TGF-β pathway is blocked to activate effector T cells and reduce the accumulation of Tregs for boosting immune response, which was further strengthened by the addition of a PD-1 blockade. Nano-IR-SB@Lip could be accumulated and deeply penetrated into the tumor tissues due to the properties of IR780. Nano-IRSB@Lip + Laser + anti-PD-1 achieved the highest response and greatly suppressed the growth of both primary and distant tumors in 4T1 tumor-bearing mice. Wang et al. utilized an anti-PD-L1 peptide (APP) and bonded it with IR780 to obtain the self-assembled IR780-M-APP nanoparticles with a precise APP loading of 48.4 wt% [76]. IR780 was able to trigger immunogenic cell death (ICD) via photodynamic effect, while the APP moiety functioned to block the PD-1/PD-L1 pathway. In vivo studies revealed that R780-M-APP NPs could successfully ablate both primary and abscopal tumors, further inhibiting lung metastasis in B16F10-bearing mice models. The porphyrin-based organic molecule has the ability to absorb light, good biocompatibility, and fluorescence imaging with NIR irradiation. Cao et al. developed self-assembled porphyrin derivatives (PPor) NPs featuring a high photothermal conversion efficiency to realize NIR-II fluorescence imaging-guided PTT [77]. Moreover, damage-associated molecular patterns (DAMPs) and tumor-associated antigens (TAAs) would be released from tumor cells under PTT to enhance immune response. The addition of PD-1 antibody to PPor NPs generated a robust antitumor activity to reduce primary and abscopal tumor growth and prevent distant growth in a breast cancer model. This work showed a potential therapeutic choice for using porphyrin derivatives as photosensitizers to treat cancers.

### 3.6. Multifunctional Nanoparticles Containing PD-1/PD-L1 Blocking

Tumor immunotherapy involves a continued multilink process including antigen presentation (Phase I) lymphocyte activation and proliferation/differentiation (Phase II), and tumor elimination (Phase III). Thus, the ideal platform is capable of simultaneously performing the three phases. Li et al. reported a three-in-one immunotherapy nanoplatform aPD-L1@HC/PM NPs containing anti-PD-L1 antibodies, chlorin e6 (Ce6)-conjugated hyaluronic acid (HC), and 1-mt-conjugated polylysine (PM) [78]. Immunogenic cell death induced by Ce6 under NIR irradiation promoted antigens releasing and initiated phase I antitumor response. The IDO inhibitor 1-mt and anti-PD-L1 antibody towards their own targets unleashed the immune-suppressive states to eliminate tumors, corresponding to the Phase II and III stages, respectively. Flow cytometry and microscopic photograph results both showed that aPD-L1@HC/PM NP plus irradiation could promote DC maturation, indicating its efficient role in PDT-induced antigen presentation for DC maturation (Phase I). In vitro studies also showed that the T cell proliferation index (PI) for the aPD-L1@HC/PM NP group was higher than that of the control group (45.6 vs. 2.66%). In addition, aPD-L1@HC/PM NP induced several cytokines, including IL-2, IFN-γ and TNF-α in Phase III. In melanoma-bearing mice models, the aPD-L1@HC/PM NPs were proven to be the most effective for abscopal tumors. Overall, this platform was valuable for boosting all-immunity-phase immunotherapy as a nanoweapon.

Chemotherapy and PTT have been combined to provoke immunogenic cell death and enhance the antitumor response of a PD-1/PD-L1 blockade in various cancer types. However, their monomodal combination was subject to an incomplete therapeutic response due to the limited accumulation of T cells. Thus, novel methods with multi-functions are urgently needed for elevating the efficacy of immunotherapy. Geng et al. constructed a core-shell metal iron–drug NPs-combining chemotherapy, PTT, and anti-PD-1 blockade [79]. This NP was generated by self-assembled Mn2+ ions, doxorubicin (DOX), and photosensitizer chlorin e6 (Ce6), then encapsulated by the red blood cell membrane to improve dispersity and stability. The coated NPs showed the highest inhibition of tumor growth in both primary and distant tumors. Moreover, Sun et al. developed a polypyrrole NP using near-infrared dye IRDye800CW and camptothecin, irradiation provided a strong PTT effect and facilitated the release of camptothecin [80]. Combining with anti-PD-L1 therapy synergistically enhanced the immune response in 4T1 mice models, eliminating primary tumor and preventing metastasis. This work offered valuable clues for combining PTT, chemotherapy and immunotherapy for the management of cancer. Table 1 summarizes novel nanoplatforms including PDT, PTT and multifunctional NPs.

## 4. Challenges and Future Perspectives

As a variety of tumor cells express inhibitory ligands to escape immune surveillance, the strategies of blocking inhibitory checkpoints, especially the PD-1/PD-L1 axis, represent a giant success in the history of cancer treatment. Nanotechnology has been introduced into cancer immunotherapy and gives us a fresh perspective. In this review, we summarized recent advances for nanoparticles-based platforms targeting the PD-1/PD-L1 pathway and the rationale for combination with other therapies. Notably, there are other emerging anticancer approaches that are not presented here, such as sonodynamic therapy (SDT) and nanovaccine [81,82,83,84]. SDT employs ultrasound to activate sonosensitizers to induce different reactions of tumor cells, such as apoptosis, necrosis, and immune-related death, including immunogenic cell death and necroptosis. Nanovaccines with specific properties consist of nanomaterials and antigens. The antigens loaded by the nanomaterials are protected from degradation during in vivo circulation and transportation. Antigen-presenting cells (APCs) targeting would be realized by decoration with specific ligands or antibodies, and cell-penetrating materials could enhance the uptake into APCs. Thus, these two strategies could activate the tumor-specific immune responses and combining them with immunotherapy would possibly elicit strong anticancer efficacy.

Beyond the PD-1/PD-L1 pathway, other immune-modulating agents are also being designed and encapsulated into nanoparticles, which could be simply divided into generalized immune therapies and personalized immune therapies [85]. Generalized immune therapies feature the improvement of the overall fitness of anticancer cells and the initiation of the killing of cancer cells previously spared by host immune response. In addition, they do not require the knowledge of individual-specific immune characteristics or biological features of the tumor. Representations of generalized immune-boosting strategies are checkpoint inhibitors, cytokines, and immune cell depletion. In contrast, personalized immune therapies aim to redirect immune cells to individual-specific tumor vulnerabilities. Information about patients must be collected to analyze tumor characteristics and immunological properties, which means significant variables like tumor mutations, TME status, and tumor-specific antigens should be evaluated before therapy is started. Nanotechnology combining any of these methods can overcome some of the challenges that limit cancer immunotherapy.

The clinical success of immunotherapy and huge advances in nanotechnology have provided substantial momentum for improving cancer treatments. Given the bottlenecks immunotherapy meets, nanomaterials have great potential to offer solutions for overcoming the limitations of immunotherapy. Researchers have incorporated PD-1/PD-L1 blockades into other combination therapies, which expanded the applicable population and greatly improve its clinical efficacy. Mechanically, nanotechnology could improve the on-target delivery to tumor sites, enhancing tumor suppression and reducing systemic adverse events. Combining ICIs with NPs encapsuling various kinds of drugs might provide a new range of dosages or combinatorial treatments. From a therapeutic point of view, there also exist some other vital questions and underlying pitfalls that should be specially addressed. First of all, more research is needed to understand the precise interactions between NPs and immune systems and the vulnerabilities of tumor cells and the best combinations for immune-based therapy remain to be elucidated. Moreover, in order to verify the efficacy of non-toxic nanoparticles combining immunotherapy and nanotechnology in humans, it is necessary to develop predictive and prognostic biomarkers and elucidate their resistance mechanisms [86]. As studies for combining cancer immunotherapy and nanoparticular approaches are quite complicated and even fail initially, researchers are supposed to optimize the in vitro and in vivo preclinical models to promote efficacy testing and to evaluate non-invasive biomarkers for early antitumor responses clinically.

As discussed in this review, nanomedicines are ideally suited for realizing effective immunotherapy in many ways. First-generation nanomedicines are constantly being improved to optimize the more advanced nanoparticles. In our view, the potential for paradigm-shifting probably depends on truly high-efficiency and low-toxic nanomedicines. However, with the preclinical evidence gradually been accumulated, given the relative sparse clinical validation, further research should focus more on their clinical translation to fill the gap. Part of the reason is that the low scalability and reproducibility of nanotechnology limits their progression in the clinic. Only a small amount of the drugs could accumulate at the tumor site, which means there lacks sufficient exact evidence on the EPR effect. The number of nanodrugs with clinical benefits is drastically lower than estimated, indicating a transformation gap of physiology and pathology which influence the behavior and functionality of nanomedicines between animal and human body. Moreover, heterogeneity between different types of cancers and cancer patients limit the development of nanomedicine, few of the approved nanodrugs are listed as first-line therapy. Consequently, in-depth studies on preclinical animal models that can highly simulate human physiological environment to establish appropriate platforms may promote the transformation of nanotechnology. The present efficacy evaluation may not be applicable for nanotechnology targeting immune systems. Thus, a more profound system should also be established to prevent early termination and monitor tumor progression.

## 5. Conclusions

In the past two decades since the PD-1 gene was discovered, PD-1/PD-L1 inhibitors have gradually played a leading role in cancer immunotherapy. At the same time, in order to achieve the precise release of the content, improve the therapeutic effect, and reduce side effects, nanoplatforms are being designed and developed to transport anti-PD-1/PD-L1 antibiotics alone or combined with chemotherapy, radiotherapy, other checkpoint inhibitors, PDT, PTT, etc. However, obstacles still exist, such as whether the incidence of adverse events is increased, despite improving the clinical efficacy, and whether drug resistance in patients who are not sensitive to immunotherapy can be effectively overcome. The targeting of nanoparticles is limited and only a few nanomaterials can be transformed for clinical practice due to the heterogeneity and biological properties of tumors, and further exploration of tumor biomarkers and intrinsic biology is needed. In conclusion, although PD-1/PD-L1 blockades have a bright future in the field of cancer immunotherapy, nanoplatforms targeting the PD-1/PD-L1 pathway still need to move forward to eventually improve the therapeutic outcomes in patients.

## Figures and Tables

**Figure 1 pharmaceutics-14-01581-f001:**
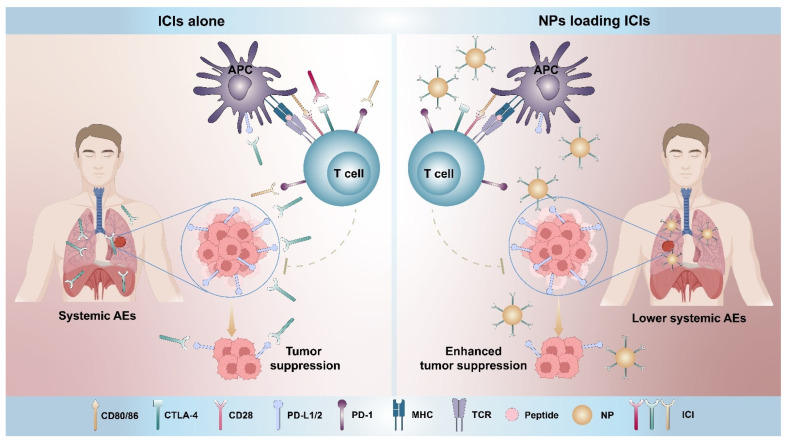
A comparison of ICIs alone and NPs loading ICIs. The application of ICIs leads to the activation of tumor-specific T cells by blocking PD-1 and CTLA-4 on T cells and also PD-L1/2 on tumor cells or APCs as well. The administration of ICIs alone over-stimulates the immune system and increases the possibility of off-target effects which is the cause of systemic AEs. By contrast, NPs loading ICIs leads to a high local concentration of ICIs in the tumor site while reducing the off-target side effects.

**Figure 2 pharmaceutics-14-01581-f002:**
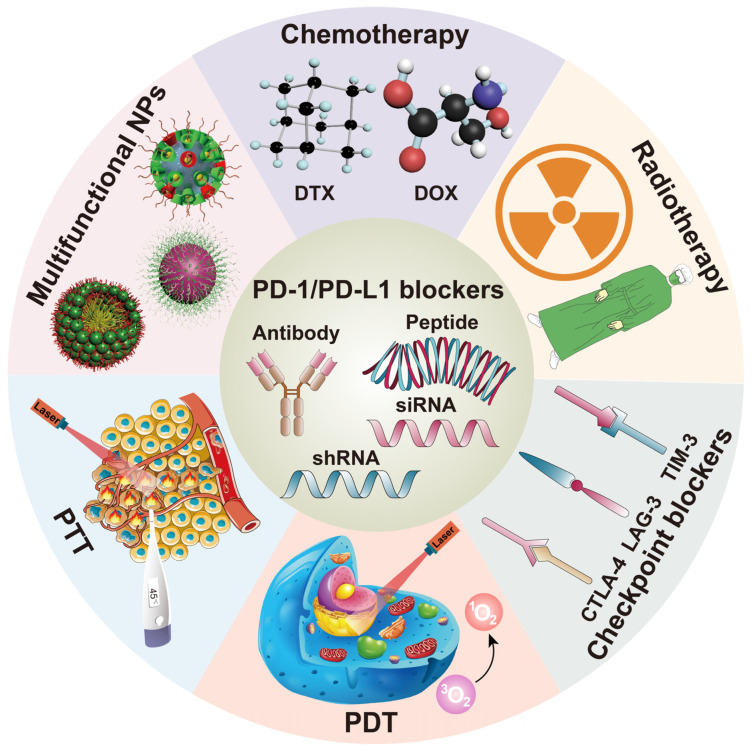
A scheme to illustrate the rationale strategies combining PD-1/PD-L1 blockers with other therapies using nanoplatforms. PDT, photodynamic therapy; PTT, photothermal therapy; NPs, nanoparticles.

**Table 1 pharmaceutics-14-01581-t001:** Summary of novel nanoplatforms including PDT, PTT and multifunctional NPs.

Main Type	Drug Delivery System	Payload	Assisted Ingredient	Responsive Condition	Ref.
αPD-1/PD-L1 +photodynamic therapy	Fe-TBP	-	αPD-L1	-	[39]
	PGCA nanoparticles	pheophorbide A	αPD-1	pH sensitive	[46]
	ZnP nanoparticles	pyrolipid	αPD-L1	-	[47]
	PcN4/albumin complexes	AQ4N	αPD-L1	-	[48]
	cell membrane nanovesicles expressing PD-1	DVDMS and PFTBA	-	-	[49]
	MMP-2-sensitive nanoparticles	indocyanine green and αPD-L1	-	MMP-2 sensitive	[55]
	lipid-based micellar nanoparticles	chlorin e6 and αPD-L1	-	pH and MMP-2 dual sensitive	[56]
	Chloringlobulin nanocomplexes	chlorin e6 and αPD-L1	-	-	[57]
αPD-1/PD-L1 +photothermal therapy	PLGA-PEG-GRGDS copolymers	αPD-1, iron oxide, and perfluoropentane	-	-	[60]
	Lipid gels	IR820 and αPD-L1	-	-	[61]
	mSiO_2_-PFP-PEG nanoparticles	copper sulfide	αPD-1	-	[63]
	DLMSNs	AUNP-12, copper sulfide, and R848	-	pH sensitive	[64]
	The porous nanocarbon	GOx	αPD-1	-	[66]
	rGO nanosheets	IDO inhibitors	αPD-L1	-	[69]
	FePS3-based nanosheets coated with the CT26 cell membrane	APP	-	-	[71]
	hydrogel	BPQD-CCNVs	αPD-1	-	[73]
	liposomes	IR780 and SB-505124	αPD-1	-	[75]
	nanoparticles	IR780 and APP	-	MMP-2 sensitive	[76]
	PPor nanoparticles	-	αPD-1	-	[77]
Multifunctional nanoparticles	nanoparticles	Ce6, 1-mt and αPD-L1	-	HAase sensitive	[78]
	nanoparticles	DOX and Ce6	αPD-1	-	[79]
	nanoparticles	pyrrole and IRDye800CW	αPD-L1	-	[80]

## Data Availability

Not applicable.

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
