# Peer review of "Recent Advances in Nanoparticles-Based Platforms Targeting the PD-1/PD-L1 Pathway for Cancer Treatment"

_pharmaceutics, 2022, doi:10.3390/pharmaceutics14081581_

Round 1

Reviewer 1 Report

The authors have carried out the in-depth review study regarding the nanoparticles-based platforms targeting the 2 PD-1/PD-L1 pathways for cancer treatment. The manuscript seems well written and performed extensive and rigorous literature review. The presentation of the writing and figures seems appropriate. This review will be of ample interest for the readers of the journal and researchers working in the field of cancer and molecular biology. This manuscript can well be accepted for the publication in the present journal; however some minor modifications and concerns can be addressed for the improvement.

1.      Minor editing and improvement of English language, grammatical and spelling errors need to be performed.

2.      Only two figures have been incorporated which seems quite basic. Figure 2 can be improved or another figure comprising various other molecular pathways can be incorporated.

3.      No table can be seen in the manuscript. Authors can incorporate one or two tables in manuscript summarizing the major findings in the context of review topic and their conclusions.

4.      Conclusion and outlook seems a bit brief and has the potential to get expanded more.

5.      Another section of “future perspectives” can be incorporated in combination with conclusion or separately.

Author Response

Thanks for your advice. All the changes in this manuscript are marked with yellow background.

1. We have carefully examined and revised the grammatical and spelling errors in this manuscript.
2. Figure 2 focused on the rational combination between anti-PD-1/PD-L1 therapy and other therapies, which was the main part of this review. We believed that it was clear and precise. This review focused on targeting the PD-1/PD-L1 pathway, various other molecular pathways were not the topic of this review.
3. We built a table to summarize novel nanoplatforms including PDT, PTT, and multifunctional NPs mentioned in this review.
4. We expanded the contents in section 4 and changed its title to 'Challenges and future perspectives'. We also added a paragraph of the conclusion as section 5.
5. The last two paragraphs of section 4 focused on the future perspectives of combining immunotherapy and nanotechnology.

Reviewer 2 Report

1. I recommend some illustration how nanoparticles recognize the PD-1 or PD-L1 receptor and then affect to the immune response. It is helpful to readers.

2. This is very valuable review article to whom interested in ICI development. Would you summarized the report about nanomedicine-based ICI (PD-1, PD-L1) as a Table ?

Author Response

Thanks for your constructive advice. All the changes in this manuscript are marked with yellow background.

1. As discussed in this review, we summarized the recent progress of nanoparticles-based platforms targeting the PD-1/PD-L1 pathway and divided them into two categories, nanoparticles directly targeting the PD-1/PD-L1 pathway and nanoparticles combining PD-1/PD-L1 blocking and other therapies. There are two ways for nanoparticles to recognize the PD-1/PD-L1 receptors, which are mentioned in this review. One is by loading or decorating PD-1/PD-L1 antibodies or peptides to the nanoparticles and directly targeting tumor cells or immune cells. The other is by loading siRNA or shRNA to inhibit the de novo expression of PD-1/PD-L1 genes. Both can enhance antitumor response in vivo experiments.
2. We built a table to summarize novel nanoplatforms including PDT, PTT, and multifunctional NPs mentioned in this review.

Reviewer 3 Report

Author reviewed the state of the art of the studies on nanocarriers targeting PD-1/PD-L1 axis and the opportunities that these strategies might offer in the next future. Authors also well discussed the combination of anti-PD-1/PD-L1 drugs and a number of nanomedicine-based treatments. Furthermore, Authors summarized potentiality and limits, posing also safety issues and prospects.

In my opinion, this work offers a complete range of information that allow a comprehensive and multidisciplinary vision of these new therapeutic approaches in cancer medicine.

I consider this work well organized and easy to read.

I recommend it for publication in this Journal after minor revisions.

Minor revisions:

1. Please define ICBs (or ICI?) in Figure 1 caption

2. Please correct OD-1 in line 80

3. In Figure 2 caption, please define the acronyms shown

4. Please, check the typing errors in the text (e.g. line 251)

5. If possible, please strenghten your statemts with further citations. References should be extended to a more international Authors' point of view 

Author Response

Thanks for your constructive advice. All the changes in this manuscript are marked with yellow background.

  1. We changed ICBs into ICIs in Figure 1 caption.
  2. We corrected OD-1 into PD-1 in line 80.
  3. We added the acronyms shown in Figure 2 caption.
  4. We have carefully examined and revised the grammatical and spelling errors in this manuscript.
  5. For this review, we used databases to search recent publications regarding nanoparticles-based platforms targeting the PD-1/PD-L1 pathway for cancer treatment. All references included are subject to this topic and there is no bias against regions of authors. Notably, there are other emerging anticancer approaches that are not presented, such as sonodynamic therapy (SDT), nanovaccine, etc. We briefly discussed them in section 4.